# Caregiver Burden among Caregivers of Patients with Mental Illness: A Systematic Review and Meta-Analysis

**DOI:** 10.3390/healthcare10122423

**Published:** 2022-11-30

**Authors:** Choy Qing Cham, Norhayati Ibrahim, Ching Sin Siau, Clarisse Roswini Kalaman, Meng Chuan Ho, Amira Najiha Yahya, Uma Visvalingam, Samsilah Roslan, Fairuz Nazri Abd Rahman, Kai Wei Lee

**Affiliations:** 1Center for Healthy Ageing and Wellness (H-CARE), Faculty of Health Sciences, Universiti Kebangsaan Malaysia, Kuala Lumpur 50300, Malaysia; p111693@siswa.ukm.edu.my (C.Q.C.); p111694@siswa.ukm.edu.my (C.R.K.); 2Institute of Islam Hadhari, Universiti Kebangsaan Malaysia, Bangi 43600, Malaysia; 3Center for Community Health Studies (ReaCH), Faculty of Health Sciences, Universiti Kebangsaan Malaysia, Kuala Lumpur 50300, Malaysia; chingsin.siau@ukm.edu.my; 4Centre for Pre-U Studies, UCSI University (Springhill Campus), Port Dickson 71010, Malaysia; homc@ucsiuniversity.edu.my; 5Department of Educational Psychology & Counselling, Faculty of Education, Universiti Malaya, Kuala Lumpur 50603, Malaysia; amirayahya@um.edu.my; 6Department of Psychiatry, Hospital Putrajaya, Putrajaya 62250, Malaysia; umavicknesh@gmail.com; 7Department of Foundation of Education, Faculty of Educational Studies, Universiti Putra Malaysia, Serdang 43400, Malaysia; samsilah@upm.edu.my; 8Psychiatry Department, Faculty of Medicine, Universiti Kebangsaan Malaysia, Kuala Lumpur 56000, Malaysia; fairuzn@ppukm.ukm.edu.my; 9Department of Medical Microbiology, Faculty of Medicine and Health Sciences, Universiti Putra Malaysia, Selangor 43400, Malaysia; lee_kai_wei@yahoo.com

**Keywords:** caregiver burden, prevalence, mental illness

## Abstract

Due to the increasing importance of caregivers in the treatment outcomes of patients with mental illness, this study aimed to systematically review studies investigating the former’s caregiver burden and determine its prevalence. An open search, without filters, was conducted. Articles were selected from Medline, Scopus, and PubMed from inception to 30 April 2022 using the PRISMA protocol. Subgroup analyses examined the between-group differences by study setting, measurements used, and disorder type. A total of 5034 caregivers from 23 countries were included in this review. Thirty-nine studies were included in the systematic review, and, among them, twenty-six were deemed eligible for meta-analysis. The overall pooled prevalence of caregiver burden among caregivers of individuals with mental illness was 31.67% (95% CI = 26.22–37.12). Pooled prevalence was the highest among care recipients receiving treatment in a hospital setting (36.06%; 95% CI = 22.50–49.63), followed by the community and clinic settings. Caregiver prevalence values were higher for burden measured using the Zarit Burden Interview (38.05%; 95% CI = 27.68–48.43). compared with other instruments, and for carers of care recipients with psychosis (35.88%; 95% CI = 27.03–44.72) compared with those without. Thus, targeted interventions should focus on caregivers of patients in hospital settings and with psychotic symptoms.

## 1. Introduction

Caregiving burden is defined as the stress that derives from caring for others, while caregiver burden is the sensation of emotional or physical tension felt by caregivers [1]. Such terms are multifaceted and aim to capture the concept behind the particular role of caring for people from different backgrounds with varying degrees of strength and resources, and who are dealing with illness-specific symptoms [2]. Treudley first proposed the concept in 1946, stating that caregiving has a negative impact on the caregiver’s mental health and quality of life [3]. In order to meet caregiving responsibilities, caregivers expend their physical, mental, social, and financial resources [4]. Caregiver burden is a widespread occurrence observed all around the world, with approximately 80% of caregivers experiencing it in their duties [3]. Factors such as the patient’s characteristics (e.g., female sex, a lower educational level), carers’ attributes, the relationship between the patient and carer (e.g., living in with the carer), time spent with the patient, and the nature and severity of the illness could impact the intensity of the caregiver burden [5,6].

Currently, an estimated 25% of the global population is suffering from mental health conditions, placing it as one of the most important factors contributing to the disease burden [7]. Correspondingly, around 450 million people across the world are experiencing mental or behavioral disorders [8]. Depression holds the highest proportion of 4.3% of the global burden of diseases and is among the largest single cause of mental illness worldwide [8]. Nearly 30% of people from all parts of the world suffer from mental illness in a year. With the increasing prevalence of psychiatric illnesses, more psychiatric patients are treated at home as compared to obtaining in-hospital treatment, which may unintentionally increase the burden of the caregivers of these psychiatric patients [9]. However, many informal caregivers who shoulder such responsibilities may not have had any proper training and education in providing care. Mental illness would thus have an influence not only on the person with the condition, but also on those who care for them [10]. Caregivers of patients with mental illness were found to have high rates of mental health difficulties compared to the general population [11]. While the benefits and drawbacks of providing care are not always obvious, they tend to result in high levels of caregiver stress [6].

Due to the considerable disability associated with their condition, those suffering from major mental illness become increasingly dependent on their caretakers. When a person suffers from severe mental illness, he or she experiences significant functional and role impairment, as well as employment disability. Approximately 90% of people with serious mental illnesses are assisted practically and emotionally on a daily basis by family carers [12]. However, many caregivers experience a significant amount of psychological pressure and are heavily burdened as a result of their lack of preparedness for the job as an informal caregiver [12,13,14]. The need and obligation to provide care deeply influence the carers’ health, employment, social life, and relationships, leading to feelings of unhappiness and dissatisfaction [10,15,16]. Suffering psychological distress and shouldering burdens while caring for a mentally ill relative affects not only the caregiver’s quality of life and health, but also their productivity as an individual and their ability to provide quality care for the ill relative, thereby worsening the ill relative’s health and decreasing the likelihood of recovery or health improvements [17,18].

Several reviews have looked into caregiver burden across various caregiving groups, including carers living with patients who have dementia [6,19,20,21,22,23]. These reviews [6,19,20,21,22,23] found that, in the 149 studies reviewed, carers’ quality of life was associated with burden and burnout syndrome, as dementia caregivers are isolated from society because of the nature of their caregiving responsibilities [19]. The findings also showed that there is a great need within this population for interventions that are successful at lowering the burden and depression symptoms [19,20]. Another review on caregivers of dementia patients reported that female caregivers tend to be more burdened than male caregivers as females undertake a greater amount of caregiving work [23]. Other reviews were conducted on carers living with older relatives [24,25,26] and carers of cancer patients [27,28]. A number of meta-analyses have also studied the association between caregiver burden and psychological distress, such as depression, anxiety, and stress [29,30,31]. Del-Pino-Casado et al. [29] reported on the association between subjective caregiver burden and symptoms of anxiety across caring groups in accordance with the COSMOS-E guide’s recommendations. Del-Pino-Casado et al. [30] examined the relationship between the subjective caregiver load of caregivers of elderly relatives and depressive symptoms. Park and Park [31] analyzed the impact of family support programs on the caregiving burden, depression, and stress experienced by family caregivers of dementia patients. Others have focused on the effectiveness of interventions to ameliorate caregiver burden among carers [32,33]. Although there are several reviews and meta-analyses on caregiver burden across different populations, especially in the area of dementia or neurocognitive disorders, there have not been any studies investigating the prevalence of caregiver burden among caregivers of individuals with mental illnesses. Data surrounding this area are thus lacking, warranting further research.

This study aimed to provide a systematic review and meta-analysis by addressing the prevalence of burden among caregivers of individuals with mental illness; this study thus hopes to give more detailed insights and shed further light in the practice and research of this particular area. The specific questions addressed in this review were as follows:

What is the prevalence of caregiver burden among caregivers of individuals with mental illness?Does the prevalence of caregiver burden differ according to study setting, measurements used, and type of disorder?

## 2. Materials and Methods

### 2.1. Protocol

The present review was conducted in accordance with the Preferred Reporting Items for Systematic Reviews and Meta-Analyses (PRISMA) and was registered with PROSPERO (CRD42021255863).

### 2.2. Design

The present study was a systematic review and meta-analysis of quantitative studies on caregiver burden among caregivers of patients with mental illness.

### 2.3. Literature Search

In compiling articles for this study, the electronic databases Medline, Scopus, and PubMed were used. The articles were selected regardless of publication date and study location, so long as it was published before or on 30 April 2022. An open search, without filters, was conducted to maximize sensitivity. A combination of search terms was used to look for relevant studies, i.e., burden* OR caregiver burden* OR burnout OR exhaustion OR strain OR overload* OR frustrate* OR stress AND caregiver* OR informal caregiver OR family caregiver OR carer* AND mental* OR mental illness OR mental disorder OR mental issues OR schizophrenia OR psychiatric disorder OR psych* problem OR mental health OR bipolar disorder OR substance abuse OR psychiatric illness OR depression OR anxiety OR psychotic disorder OR obsessive-compulsive disorder OR behaviour disorder OR behavioural disorder. Forward and backward searches were conducted as well to identify suitable articles. The search strategies are shown in Table 1.

### 2.4. Study Selection

Firstly, relevant articles identified through the databases were imported into the Endnote program X5 version and any duplicate publications were removed. This step was performed by two investigators (C.Q.C. and C.R.K.) independently. Secondly, the two investigators (C.Q.C. and C.R.K.) independently screened the titles and abstracts to ensure the eligibility of the articles. Thirdly, full-text articles were assessed independently by the same two investigators, based on the inclusion criteria mentioned below. Any disagreements that arose were resolved through a discussion before the commencement of the quantitative analysis.

### 2.5. Inclusion Criteria

Cohort studies and cross-sectional studies were eligible for quantitative analysis if the participants in the study were the caregivers of individuals with mental illness. The studies must also have been published in an English peer-reviewed journal. Any articles that reported on the percentage, prevalence, or mean score of burden with any screening or diagnostic tools were included in this study. The disorders that are listed in the Diagnostic and Statistical Manual of Mental Disorders, Fifth Edition [34] were included in the study. Studies on caregivers of patients with mental illness, in general, were also included.

### 2.6. Exclusion Criteria

Based on the authors’ consensus, we decided to exclude major neurocognitive disorders or dementia, as a recent meta-analytic study had addressed this condition [21]. If the first and second investigators (C.Q.C. and C.R.K.) had doubts concerning the eligibility of a study to be included, then the third and fourth reviewers (K.W.L. and C.S.S.) were involved in the decision-making process. We also excluded pilot, qualitative, validation, psychometric, preliminary, randomized controlled trial, systematic review, meta-analysis, treatment-related, and interventional studies. Unpublished reports, seminar presentations, doctoral and master’s theses, and grey literature searches were not included in the study.

### 2.7. Data Extraction

The data were extracted and managed in a pre-designed form in Microsoft Excel. A form listing the name of the first author, publication year, study location, study design, study setting, sample size, and measuring instruments was then used to assess article characteristics. Data extraction was conducted independently by three reviewers (C.Q.C., C.S.S., and C.R.K.), and the results of the data extraction were compared between the three reviewers to ensure that there were no errors.

### 2.8. Quality Assessment

The quality of the included articles was assessed based on the Strengthening the Reporting of Observational Studies in Epidemiology (STROBE) checklist [35]. The results of the assessment are shown in Appendix A. There are 22 proposed items in the checklist, with items 6, 12, 14, and 15 having specific variations, which assessed 6 components for cohort, case–control, and cross-sectional studies. The absence or presence of a component stated in each item from the article was graded with a “0” or a “1”, respectively. Each article was graded as “low risk of bias” if the STROBE score was ≥14/22; or “high risk of bias” if the score was <14/22. Two investigators (C.Q.C and C.R.K.) individually assessed the study’s quality, and any discrepancies were resolved by discussion with the third investigator (C.S.S.). Fifteen studies scored ≥14 and were graded as having “low risk of bias”, while 24 studies scored ≤14 and were graded as having “high risk of bias”. Studies were nevertheless included in the analysis regardless of the STROBE score and grading (refer to Appendix A).

### 2.9. Statistical Analysis

A meta-analysis using a continuous random-effect model (DerSimonian and Laird method) was performed. Random-effects meta-analyses allow for heterogeneity by assuming that underlying effects follow a normal distribution, but they must be interpreted carefully. Heterogeneity between the trials was assessed by considering the I^2^ inconsistency statistic. An I^2^ estimate greater than or equal to 70% was interpreted as an indication of substantial levels of heterogeneity. If the quantitative analysis was unsuitable due to the heterogeneity and/or a small number of studies, a narrative overview of the findings of the included studies was presented with tabular summaries of extracted data.

Since the mean of burden was available from 26 of the articles, the mean of burden was used to conduct the meta-analysis of this study to estimate the pooled prevalence of caregiver burden [36,37]. Percentages and 95% confidence intervals were used as summary statistics for the pooled prevalence. Sensitivity analyses were used to examine whether overall findings were robust to potentially influential decisions. Prediction intervals from random-effects meta-analyses are a useful device for presenting the extent of between-study variation. The data from each study (care recipients, country, study design, study setting, and measures for burden) were used to build tables for an overall description of the included studies. As study populations and data sources differed among the included studies, an evaluation of studies was conducted to determine whether they were suitable for meta-analysis, and only suitable studies were included in the quantitative analysis. Open Meta (Analyst) [38] was used to conduct the meta-analysis for this study.

### 2.10. Subgroup Analyses

Subgroup analyses are useful to examine the between-group differences in terms of the prevalence as a possible cause of heterogeneity across studies. The prevalence of caregivers of individuals with mental illness was examined by subgrouping the study setting, measurements used, and type of disorder. The prevalence of caregiver burden was reported in percentages with a 95% confidence interval (CI).

### 2.11. Sensitivity Analysis

We performed sensitivity analysis by using the leave-one-out meta-analysis to examine how each particular study altered the overall performance of the rest of the studies, especially with regard to the pooled prevalence estimates and heterogeneity. We used the I^2^ (Higgins et al. [39]) to measure the proportion of heterogeneity due to the variability of effect estimates amongst individual studies, with values of 25, 50, and 75% indicative of mild, moderate, and severe heterogeneity, respectively.

## 3. Results

### 3.1. Description of Included Studies

A total of 4983 articles were identified in the initial screening. After removing the duplicate articles (*n* = 598), 4385 articles were retrieved for further assessment. After screening for suitability based on the title and abstract, 4341 articles were excluded, and 44 articles were selected for full-text assessment. After a thorough evaluation, a total of 27 articles were identified to be suitable to be included in the systematic review. Another 52 articles were identified through forward and backward searches. However, of these 52 articles, only 12 articles were deemed suitable to be included in the systematic review. Therefore, a total of 39 articles were included in the systematic review. Among the 39 articles, 26 articles were deemed eligible for meta-analysis (refer to Figure 1).

The main characteristics of the included studies (*n* = 39) were tabulated as shown in Appendix A. Among the included studies, 23 studies used a cross-sectional design. A total of 5034 caregivers from 23 countries were included in the analysis. Six studies were conducted in Nigeria [18,40,41,42,43,44], five studies in Brazil [45,46,47,48,49], three studies each in Nepal [50,51,52] and Turkey [53,54,55], two studies each in Hong Kong [56,57], the USA [58,59], and Taiwan [60,61], and one study each in Africa [62], China [63], Chile [64], Egypt [65], Greece [66], India [67], Ireland [68], Italy [69], Japan [70], Jordan [71], Kuwait [72], Poland [73], Portugal [74], Singapore [75], Spain [76], and the Netherlands [77]. In terms of study setting, 11 studies were conducted in the community, 18 studies in the hospital, and nine studies in the clinic (refer to Appendix A).

With regard to the care recipients of the caregivers, 12 studies were conducted on patients with mental illness in general, 12 studies focused on patients with schizophrenia, while five studies focused on bipolar affective disorders with schizophrenia-related disorders. Two studies focused on patients with depressive disorders. One study each was conducted on patients who were suffering from autism spectrum disorders, down syndrome, neurodevelopmental disorders, neuropsychiatric illness, minor psychiatric disorders, and obsessive-compulsive disorders.

The most used instrument to measure caregiver burden was the Zarit Burden Interview [78], which was used in 17 of the studies. Three studies each used the Family Burden Interview Schedule [79] and the Involvement Evaluation [80]. Two studies each employed the Caregiver Strain Index [81] and Burden Assessment Scale [82]. Other studies, on the other hand, utilized the Caregiver Burden Inventory [83], Family Burden Scale [84], Family Problems Questionnaire [85], Feetham Family Functioning Scale [86], Perceived Chronic Strains Scale [87], Self-Perceived Pressure by Informal Care Scale [88], Social Behavior Assessment Schedule [89], Burden Assessment Schedule [90], Burden Questionnaire [91], and Caregiver Burden Scale [92]. In two of the studies [45,50], the researchers constructed their own questionnaire to measure caregiver burden.

### 3.2. Pooled Prevalence of Caregiver Burden among Caregivers of Patients with Mental Illness

A summary of the pooled prevalence of caregiver burden among caregivers of patients with mental illness is shown in Figure 2. The pooled prevalence was conducted on the 26 articles that were deemed eligible for meta-analysis. The overall pooled prevalence was 31.67% (95% CI = 26.22–37.12). The highest prevalence of caregiver burden recorded was 86.51% (95% CI = 82.54–90.48), while the lowest prevalence was 3.13% (95% CI = 2.36–3.90). More than half of the studies (*n* = 14) recorded a prevalence of caregiver burden below the average prevalence of 30%, while twelve studies recorded a prevalence of caregiver burden above the average (refer to Figure 2).

### 3.3. Subgroup Analysis for the Prevalence of Caregiver Burden among Caregivers of Patients with Mental Illness According to Study Setting, Instruments, and Type of Mental Illness

Table 2 summarizes the subgroup analysis of the pooled prevalence of caregiver burden among caregivers of patients with mental illness according to study setting and instruments used, as well as the disorders. Forest plots for study setting, the instruments used, and the disorders are shown in Figure 3, Figure 4, and Figure 5, respectively (refer to Figure 3, Figure 4 and Figure 5). The pooled prevalence of caregiver burden was highest among the care recipients who received treatment in a hospital setting (36.06%; 95% CI = 22.50–49.63), followed by the community setting (28.28%; 95% CI = 18.97–37.58), while the lowest pooled prevalence of caregiver burden among the care recipients who received treatment was in a clinic setting (27.52%; 95% CI = 14.78–40.26). Subgroup analysis according to the instrument used showed that the highest burden of care was were reported in studies using the Zarit Burden Interview (36.90%; 95% CI = 28.17–45.62). In terms of the type of disorder, the pooled prevalence of caregiver burden was highest among the caregivers of care recipients who suffered from psychotic disorders (35.88%; 95% CI = 27.03–44.72) (refer to Table 2).

## 4. Discussion

To the best of our knowledge, this review is the first systematic review and meta-analysis on the caregiver burden of carers for individuals with mental illness. We aimed to investigate the prevalence of burden among caregivers of individuals with mental illness. A total of 5034 caregivers from 23 countries were included in the analysis. Thirty-nine articles were included in the systematic review and 26 articles were deemed eligible for meta-analysis. Subgroup comparisons across study settings, the measurement used, and the type of mental illness were conducted. The main finding of this study was that the overall pooled prevalence of caregiver burden among caregivers of individuals with mental illness was 31.67%. Subgroup analyses showed that caregivers in hospital settings (36.06%), studies using the Zarit Burden Interview (36.90%), and caregivers of individuals suffering from psychosis (35.88%) recorded significantly higher prevalence values.

In this study, we found that nearly one third of caregivers of individuals with mental illness, excluding major cognitive disorders, suffered from caregiver burden. The caregiver burden prevalence of 31.67%% found in our study, however, is lower than the prevalence of caregiver burden found in other meta-analytic studies [20,93] which cover a wide range of conditions, including physical and mental illnesses. For example, the pooled prevalence of caregiver burden among caregivers of dementia patients was nearly twice as high, at 49.26% [20]. On the other hand, a meta-analysis conducted in Iran on the caregiver burden of carers for chronic illness patients showed that more than half (53.28%) had caregiver burden, and the prevalence of caregiver burden for mentally ill patients (58.7%) was comparable to that of Alzheimer’s patients (57.1%) [93]. A meta-analysis that compared the caregiver burden of carers for physical vs. mental illnesses showed that carers for patients with physical illnesses recorded significantly lower caregiver burden mean scores than those with cognitive impairment or dementia, Alzheimer’s, and mental illnesses [94]. As the heterogeneity between studies included in this meta-analysis was high, there may be a need to examine further the factors that contributed to the high variances, and the reasons contributing to the lower prevalence.

This study also found that carers for mentally ill individuals within the schizophrenia spectrum disorder or with psychosis recorded a higher caregiver burden prevalence (35.88%) than for carers in studies that did not mention the presence of psychosis in the patients (26.82%). Caregivers who care for patients with psychotic symptoms face a greater burden than those who care for patients with bipolar disorders, with a higher burden reported by laborers and housewives [95]. Symptoms of psychosis, such as disorganized thoughts, hallucinations, and delusions, may require constant supervision to ensure the patient’s personal hygiene and grooming and prevent the patient from engaging in negative behaviors such as skipping medication [95,96]. The greater burden felt by caregivers of schizophrenia patients may also be due to the need for caregiving even during remission, and the social exclusion experienced by the caregivers or patients [95]. The caregiver burden of schizophrenia patients was higher among older and unemployed individuals, mothers, those reporting lower educational levels, and caretakers of younger patients [64]. Moreover, a study further found that the caregiver burden may be due to higher psychological morbidity and maladaptive coping in caregivers or schizophrenia patients [97]. Another study reported that, compared to carers of patients with depression, a significantly higher percentage of carers of patients with schizophrenia reported worrying about the future and finances of the patient [98]. Carers of schizophrenia patients also reported providing more motivation and encouragement to schizophrenia patients in the latter’s care [98]. Greater worry and the provision of nursing care may have contributed to a greater burden among carers for patients with psychosis or schizophrenia spectrum disorders. Moreover, the higher stigma against individuals with schizophrenia vs. depression or other mental illnesses, particularly in terms of perceptions of dangerousness and negative stereotyping for schizophrenia patients [99,100], may further isolate the caregiver from sources of social support.

Caregivers of individuals with mental illness in hospital settings in this study reported a higher caregiver burden compared with those in clinic and community settings. The characteristics of the patients requiring hospitalization may indicate a need for greater care, such as a more severe presentation of the mental illness symptoms or exhibiting self-harm or suicidal behaviors [101,102]. A study found that patients who were involuntarily admitted to the hospital had a higher likelihood of more severe psychotic symptoms, aggressive behavior, and medication non-adherence [101]. Caregiver burden may be a factor determining the decision to institutionalize older people in hospital settings [103]. Limited engagement with the healthcare system due to a lack of health literacy, perceptions of ineffective healthcare provided, and limited access to healthcare services has been associated with greater caregiver burden [2]. In another study on caregiver burden among carers of children with neurodevelopmental disabilities, caregivers experienced a greater burden when they found it difficult to access and navigate within the healthcare system or reported unmet healthcare needs [104]. Therefore, caregivers of hospital-dwelling patients may be experiencing a higher caregiver burden as they may lack the support provided by the healthcare system.

The Zarit Burden Interview (ZBI) is the most commonly used instrument to measure caregiver burden. The findings are consistent with a review of instruments measuring caregiver burden for mental illness patients conducted by Schulze and Rössler [105]. The other instruments used to measure caregiver burden are the Family Burden Interview Schedule [80], Involvement Evaluation [80], Caregiver Strain Index [81], Burden Assessment Scale [82], Caregiver Burden Inventory [83], Family Burden Scale [84], Family Problems Questionnaire [85], Feetham Family Functioning Scale [86], Perceived Chronic Strains Scale [87], Self-Perceived Pressure by Informal Care Scale [88], Social Behavior Assessment Schedule [89], Burden Assessment Schedule [90], Burden Questionnaire [91], and Caregiver Burden Scale [92].

Despite the use of a number of validated questionnaires, two of the studies reviewed used a self-designed or adapted questionnaire [45,50]. It was interesting to note that the caregiver burden prevalence using the Zarit Burden Interview was higher (36.90%) in comparison with studies utilizing other instruments (26.47%). The difference may be attributed to the fact that these questionnaires were originally developed for measuring caregiver burden among different populations, and they have different factor structures and target populations. For example, the ZBI was developed with the purpose of measuring the burden of caregivers for dementia patients and is now considered the gold standard and a generic measurement for caregiver burden for diverse diseases. Meanwhile, the Involvement Evaluation Questionnaire was used more specifically to assess the caregiver burden among those caring for patients with severe mental illnesses such as schizophrenia [74]. The Caregiver Burden Inventory, on the other hand, provides a multidimensional view of caregiver burden (time dependence, developmental, physical, social, and emotional burden) and may be useful in providing specific areas for intervening with caregiver burden based on these dimensions [83]. Future studies estimating the prevalence of caregiver burden should specify which questionnaire they are using and note the tendency for the ZBI to yield a higher estimated prevalence of caregiver burden.

More than half (62%) of the included studies in our review had a high risk of bias. There were 24 articles with a “high risk of bias”, while the remaining studies were graded as having a “low risk of bias” (*n* = 15). Items in the STROBE checklist, such as “use of a flow diagram”, “sources of bias”, and “sample size calculation”, were not commonly reported. Two studies used self-designed questionnaires to measure caregiver burden, which may have resulted in limited comparison with studies using commonly used scales such as the Zarit Burden Interview. As a result of the high risk of bias of 24 studies and variations in the caregiver burden measurement tools, only a small selection of research could be included for meta-analysis. This research has some limitations. The search technique was confined to peer-reviewed and published articles in international databases. Unpublished reports, seminar presentations, doctoral and master’s theses, and grey literature searches were not included in the study. The pre-specified criteria for this study may be too narrow, resulting in the exclusion of potentially relevant studies from our analysis. All the research considered in this review used cross-sectional designs, which prevented causal conclusions from being drawn. The use of several scales to measure caregiver burden may have contributed to the higher heterogeneity between studies. In addition, we found high levels of within-group heterogeneity to be present among subgroups, indicating that these groups may not account for the variance between studies and that the results of subgroup analyses may need to be interpreted with caution because of uneven covariate distributions among groups. As a result, it would be worthwhile to conduct additional studies to address these constraints.

## 5. Implications

The results of this study support a renewed emphasis on interventions to identify the caregiver burden for the growing number of informal caregivers. The findings of this study have drawn attention to the possibility that caregivers of individuals with mental illness may require psychological help in order to cope with the burden that they face. By taking care of loved ones at home, caregivers significantly contribute to the reduction of expenses and resources for the healthcare system. Therefore, it is essential to provide a support framework to lessen the burden on caregivers. There also appears to be a lack of cohort studies addressing caregiver burden, which could provide higher-quality evidence of caregiver burden across time. Researchers choosing measurement tools to measure caregiver burden should be aware that the Zarit Burden Interview may provide higher burden scores in comparison with other instruments measuring caregiver burden.

## 6. Conclusions

In conclusion, this study revealed that nearly one third of the caregivers experienced a burden when taking care of individuals with mental illness. This study suggests that the prevalence of caregiver burden was significantly higher for carers in hospital settings, studies utilizing the Zarit Burden Interview, and caregivers of individuals with psychosis. Given their increasing importance in the treatment outcomes of psychiatric patients in the age of deinstitutionalization, caregivers of people with mental illness should therefore receive more attention.

## Figures and Tables

**Figure 1 healthcare-10-02423-f001:**
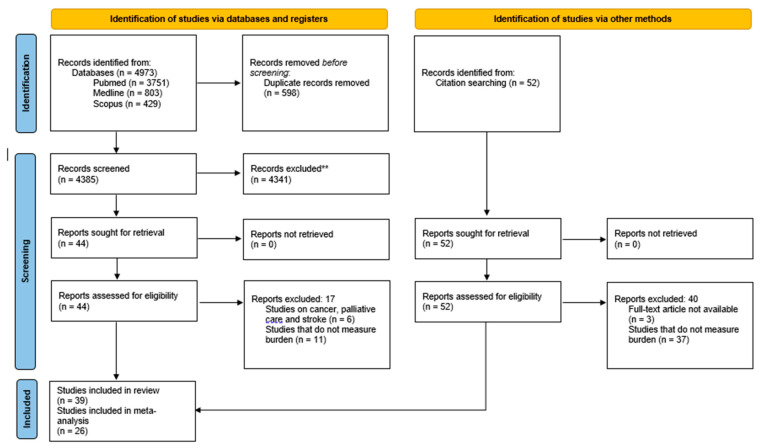
Preferred Reporting Items for Systematic Reviews and Meta-Analyses (PRISMA) flow diagram of the literature screening process.

**Figure 2 healthcare-10-02423-f002:**
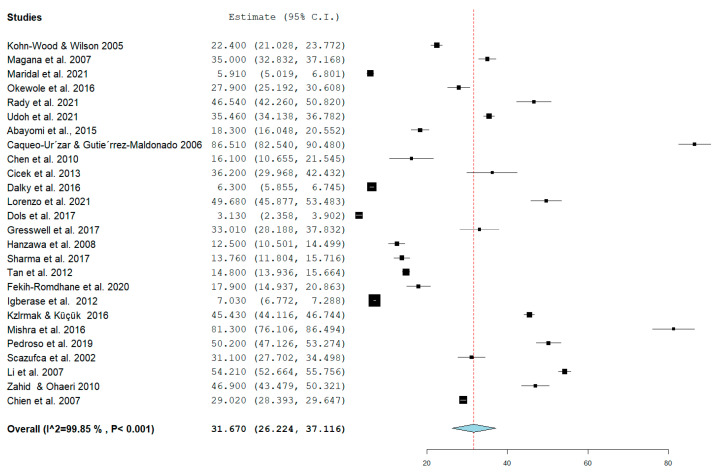
Pooled prevalence of caregiver burden among caregivers of patients with mental illness [18,40,41,44,48,49,50,51,52,54,55,56,58,59,60,62,63,64,65,68,69,70,71,72,75,77].

**Figure 3 healthcare-10-02423-f003:**
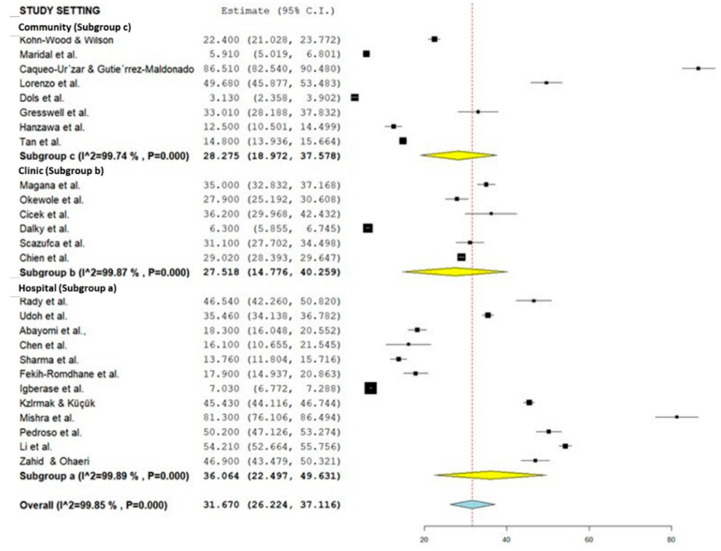
Subgroup forest plot on the study setting (Community: [50,58,64,68,69,70,75,77]; Clinic: [40,49,54,56,59,71]; Hospital: [18,41,44,48,51,52,55,60,62,63,65,72]).

**Figure 4 healthcare-10-02423-f004:**
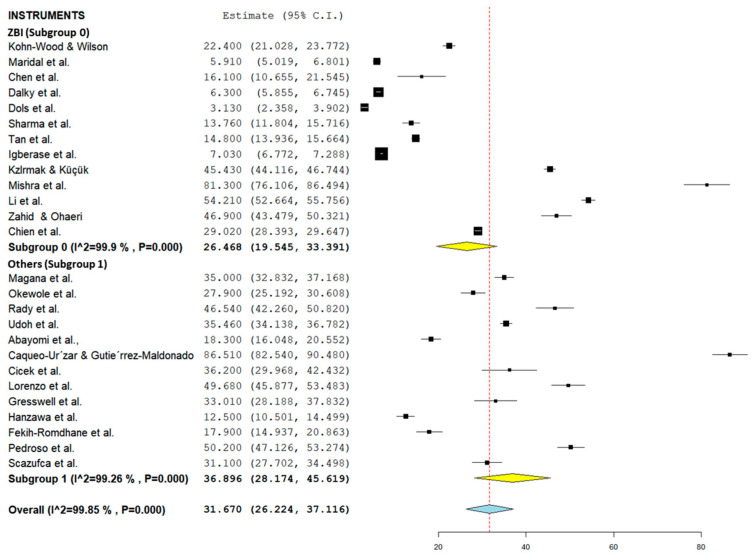
Subgroup forest plot on the types of measurement used (Zarit: [44,50,51,52,55,56,58,60,63,71,72,75,77]; Others: [6,18,40,41,48,49,54,62,64,65,68,69,70]).

**Figure 5 healthcare-10-02423-f005:**
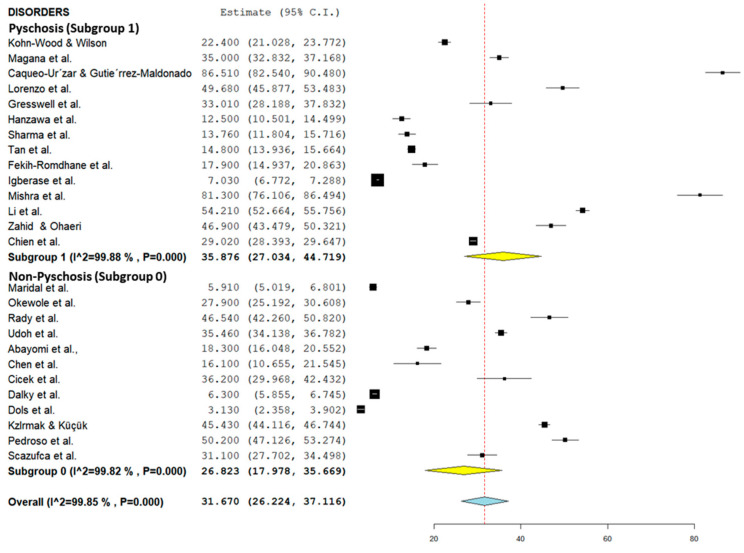
Subgroup forest plot of psychotic and non-psychotic disorders (Psychosis: [44,51,52,56,58,59,62,64,68,69,70,72,75]; Non-psychosis: [18,40,41,48,49,50,54,55,60,65,71,77]).

**Table 1 healthcare-10-02423-t001:** Search terms and strategy used in PubMed, Scopus, and MEDLINE.

(1)	burden* OR caregiver burden* OR burnout OR exhaustion OR strain OR overload* OR frustrate* OR stress
(2)	caregiver* OR informal caregiver OR family caregiver OR carer*
(3)	mental* OR mental illness OR mental disorder OR mental issues OR schizophrenia OR psychiatric disorder OR psych* problem OR mental health OR bipolar disorder OR substance abuse OR psychiatric illness OR depression OR anxiety OR psychotic disorder OR obsessive-compulsive disorder OR behaviour disorder OR behavioural disorder

*Note. truncation technique* for SCOPUS and PubMed.*

**Table 2 healthcare-10-02423-t002:** Subgroup analysis of pooled prevalence of caregiver burden among caregivers of patient with mental illness.

Variables		No. of Studies	Prevalence, %	95% CI	I^2^, %	*p*-Value
Study setting	Community (Subgroup c)	8	28.28	18.97–37.58	99.74	<0.001
	Clinic (Subgroup b)	6	27.52	14.78–40.26	99.87	<0.001
	Hospital (Subgroup a)	12	36.06	22.50–49.63	99.89	<0.001
	Subtotal	26	31.67	26.22–37.12	99.85	<0.001
Instruments	The Zarit Burden Interview (Subgroup 0)	13	36.90	28.17–45.62	99.26	<0.001
	Others (Subgroup 1)	13	26.47	19.55–33.40	99.90	<0.001
	Subtotal	26	31.67	26.22–37.12	99.85	<0.001
Disorders	With psychosis (Subgroup 1)	14	35.88	27.03–44.72	99.88	<0.001
	Without psychosis (Subgroup 0)	12	26.82	17.98–35.67	99.82	<0.001
	Subtotal	26	31.67	26.22–37.12	99.85	<0.001

## Data Availability

Not applicable.

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
