# Peer review of "Caregiver Burden among Caregivers of Patients with Mental Illness: A Systematic Review and Meta-Analysis"

_healthcare, 2022, doi:10.3390/healthcare10122423_

Round 1

Reviewer 1 Report (Previous Reviewer 1)

Dear authors, 

Thank you for making the proposed changes. The manuscript is more complete. 

Best regards,

Author Response

Point 1: Dear authors, 

Thank you for making the proposed changes. The manuscript is more complete. 

Best regards

Response 1: Thank you for the comment.

Reviewer 2 Report (New Reviewer)

This manuscript is a systematic review and meta-analysis of the caregiver burden of carers for individuals with mental illness. Overall the manuscript could use some careful editing for language.

Please find comments below: 

1) Line 52, With the increasing prevalence of psychiatric illnesses, please explain why
2) Line 87, explanations unclear, please indicate exactly for whom with psychological distress such as depression, anxiety, and stress.

2) Line 81: Again, not quite true. Need to say that "..e-cig use is ASSOCIATED with an increased risk of cig smoking..."

3) At the end of the Statistical Analysis subsection, please provide the statistical software used to complete the analysis.
4) There are too many tables and figures. The information in Table 2 can be moved into the appendix.

5) The information in figures can be merged into one mixed multiple graph with 4 plots.

 6) Please merge section 5 Implication and Section 6 Conclusion

Author Response

Response to Reviewer 2 Comment

This manuscript is a systematic review and meta-analysis of the caregiver burden of carers for individuals with mental illness. Overall the manuscript could use some careful editing for language.

Please find comments below: 

Point 1: Line 52, With the increasing prevalence of psychiatric illnesses, please explain why

Response 1: We have added a statement to clarify the increasing prevalence of psychiatric illnesses as follows:

(Introduction, p. 2, ll. 47-51) As the COVID-19 disease expanded across the globe, nations reacted by implementing nationwide lockdowns in an effort to contain and stop the virus's spread. In April 2020, with more than 50% of the world's population under some sort of lockdown, concerns over population mental health increased as people encountered previously unheard-of levels of known risk factors for mental health illness (Winkler et al., 2020).

Point 2: Line 87, explanations unclear, please indicate exactly for whom with psychological distress such as depression, anxiety, and stress.

Response 2: Thank you for the comment. The participants with psychological distress such as depression, anxiety, and stress by each articles has been explained as below:

(Introduction, p. 2, ll. 91-97) Del-Pino-Casado et al.33 reported on the association between subjective caregiver burden and symptoms of anxiety across caring groups in accordance with the COSMOS-E guide's recommendations. Del-Pino-Casado et al.34 examined the relationship between caregivers of elderly relatives' subjective caregiver load and depressive symptoms. Park and Park 35 analyzed the impact of family support programmes on the caregiving burden, depression, and stress experienced by family caregivers of dementia patients.

Point 3: Line 81: Again, not quite true. Need to say that "..e-cig use is ASSOCIATED with an increased risk of cig smoking..."

Response 3: Thank you for pointing this out. The sentence now has been changed to ‘associated with’ instead of ‘severely impacted.

(Introduction, p. 2, ll. 82) These reviews 7,20–24found that, in the 149 studies reviewed, carers' quality of life was associated with burden and burnout syndrome as dementia caregivers are isolated from society because of the nature of their caregiving responsibilities25.

Point 4: At the end of the Statistical Analysis subsection, please provide the statistical software used to complete the analysis.

Response 4: We have added the statistical software used to complete the analysis at the end of the Statistical Analysis subsection as follows:

(Statistical Analysis, p. 5, ll. 200) Open Meta (Analyst)39 was used to conduct meta-analysis for this study.

Point 5: There are too many tables and figures. The information in Table 2 can be moved into the appendix.

Response 5: Thank you for the recommendation. Table 2 have been moved to supplementary file. The table has been renamed as ‘Table S2. Characteristics of Included Studies.’

Point 6: The information in figures can be merged into one mixed multiple graph with 4 plots.

Response 6: Thank you for the recommendation. Unfortunately, we are not able to merge the figures as the analysis needed to be conducted separately. The software generates the graphs by default. However, we have summarized the information in Table 2 as below:  

Variables

No. of Studies

Prevalence, %

95% CI

I2, %

P-value

Study setting

Community

8

28.28

18.97-37.58

99.74

<0.001

Clinic

6

27.52

14.78-40.26

99.87

<0.001

Hospital

12

36.06

22.50-49.63

99.89

<0.001

Subtotal

26

31.67

26.22-37.12

99.85

<0.001

Instruments

The Zarit Burden Interview

13

36.90

28.17-45.62

99.26

<0.001

Others

13

26.47

19.55-33.40

99.90

<0.001

Subtotal

26

31.67

26.22-37.12

99.85

<0.001

Disorders

With psychosis

14

35.88

27.03-44.72

99.88

<0.001

Without psychosis

12

26.82

17.98-35.67

99.82

<0.001

Subtotal

26

31.67

26.22-37.12

99.85

<0.001

 Point 7: Please merge section 5 Implication and Section 6 Conclusion

Response 7: Thank you for the comment. We have merged the 2 sections as ‘Section 5 Conclusion’.

Reviewer 3 Report (New Reviewer)

This systematic review synthesized the prevalence of caregiver burden in patients with mental health illness. Study results provide quantitative evidence, which demonstrates the high prevalence of caregiver burden in this population. My greatest concern is the very high heterogeneity, which makes the meta-analysis results unreliable and hard to interpret. More analysis may be made to explore the source of heterogeneity. Some specific comments are as follows:

  1. Line 110: Any other sources other than three databases? Such as clinical registry and grey literature.
  2. Line 141: Were conference papers, commentary, and master/doctoral theses excluded from the review? This information should be presented clearly.
  3. Table S1: This table is hard to read. The font size is too small. May split it into two tables. Moreover, please provide the first authors’ name and publication year, along with the citation number. It may be much clearer.
  4. Line 169: I would like to know how authors calculate the prevalence of caregiver burden. Some studies may only provide a mean score of the scale rather than a percentage of participants with a high caregiver burden. Were these studies excluded from this review? Or any methods used to calculate the prevalence? Also, studies may adopt different cut-off values of the scale used for caregiver burden. How to address these issues?
  5. Line 174: Please provide citations about the criteria for a high level of heterogeneity.
  6. Figure 1: Please use the PRISMA 2020 flow diagram.
  7. Line 215: Five studies did not mention the study design. So were they excluded from this review? If not, are there any methods for addressing this issue (such as contacting the original authors for confirmation)?
  8. Line 238: Please provide citations for the two studies.
  9. Please summarize the quality of included studies in the results. The discussions should also state the key methodological limitations of the studies.
  10. Table 2: Please provide the authors’ name and publication year; the participants’ relationship with patients; and the mean age of the study participants.
  11. Table 3: All meta-analyses showed a very high heterogeneity (over 99%). Could the authors discuss the impact of the high heterogeneity on the study results?
  12. Figures 3,4, and 5: Please indicate the name of each subgroup in these figures.

Author Response

Response to Reviewer 3 Comment

This systematic review synthesized the prevalence of caregiver burden in patients with mental health illness. Study results provide quantitative evidence, which demonstrates the high prevalence of caregiver burden in this population. My greatest concern is the very high heterogeneity, which makes the meta-analysis results unreliable and hard to interpret. More analysis may be made to explore the source of heterogeneity. Some specific comments are as follows:

Point 1: Line 110: Any other sources other than three databases? Such as clinical registry and grey literature.

Response 1: Only the electronic databases Medline, Scopus, and PubMed were used. Forward and backward searches were conducted as well to identify suitable articles. Other sources such as clinical registry and grey literature were not included in the study. We have added a statement to clarify as follows:

(Exclusion Criteria, p. 4, ll. 158-160) Unpublished reports, seminar presentations, doctoral and master's theses, and grey literature searches were not included in the study.

Point 2: Line 141: Were conference papers, commentary, and master/doctoral theses excluded from the review? This information should be presented clearly.

Response 2:    Thank you for the comment. Conference papers, commentary, and master/doctoral theses were excluded from the review. We have added a statement to clarify as follows:

(Exclusion Criteria, p. 4, ll. 158-160) Unpublished reports, seminar presentations, doctoral and master's theses, and grey literature searches were not included in the study.

Point 3: Table S1: This table is hard to read. The font size is too small. May split it into two tables. Moreover, please provide the first authors’ name and publication year, along with the citation number. It may be much clearer.

Response 3: Thank you for the recommendation. The table has been split into 3 tables for a clearer view. We have also added the first authors’ name and publication year, along with the citation number, as seen in Table S1, column 2.

Point 4: Line 169: I would like to know how authors calculate the prevalence of caregiver burden. Some studies may only provide a mean score of the scale rather than a percentage of participants with a high caregiver burden. Were these studies excluded from this review? Or any methods used to calculate the prevalence? Also, studies may adopt different cut-off values of the scale used for caregiver burden. How to address these issues?

Response 4: Thank you for the comment. In the meta-analysis, only the studies that has mean score of burden were included in the meta-analysis. Hence, the prevalence of caregiver burden was derived from those articles with mean score of burden.

Point 5: Line 174: Please provide citations about the criteria for a high level of heterogeneity.

Response 5: We have added as follows:

We used the I2 (Higgins et al. 40) to measure the proportion of heterogeneity due to the variability of effect estimates amongst individual studies, with values of 25, 50 and 75% indicative of mild, moderate and severe heterogeneity, respectively.

Point 6: Figure 1: Please use the PRISMA 2020 flow diagram.

Response 6: Thank you for the comment. We have now used the PRISMA 2020 flow diagram, as seen in Figure 1.

Point 7: Line 215: Five studies did not mention the study design. So were they excluded from this review? If not, are there any methods for addressing this issue (such as contacting the original authors for confirmation)?

Response 7: These five studies were included in the review. We evaluated each article based on its methodology and inferred the study design. As long as the articles were within our inclusion criteria, we included them.

Point 8: Line 238: Please provide citations for the two studies.

Response 8: The citation of the two studies have been added.

(Results, p. 7, ll. 255) In two of the studies46,51, the researchers constructed their own questionnaire to measure caregiver burden.

Point 9: Please summarize the quality of included studies in the results. The discussions should also state the key methodological limitations of the studies.

Response 9: Thank you for pointing this out. The quality of the included studies has been summarized.

(Quality Assessment, p. 4, ll.178-179) Fifteen studies scored ≥14 and are graded as ‘good’, while 24 studies scored ≤14 and are graded as ‘poor’.

 Point 10: Table 2: Please provide the authors’ name and publication year; the participants’ relationship with patients; and the mean age of the study participants.

Response 10: Thank you for the recommendations. The authors’ name and publication year; the participants’ relationship with patients; and the mean age of the study participants have been added to the table, as seen in Table S2 (supplementary file), column 2 (authors’ name and publication year), column 9 (mean age of the study participants) and column 10 (the participants’ relationship with patients).

Point 11: Table 3: All meta-analyses showed a very high heterogeneity (over 99%). Could the authors discuss the impact of the high heterogeneity on the study results?

Response 11: Thank you for the comment. the impact of the high heterogeneity on the study results has been discussed as below:

(Discussion, p.5, ll. 462-466) In addition, we found high levels of within-group heterogeneity to be present among subgroups, indicating that these groups may not account for the variance between studies and that the results of subgroup analyses may need to be interpreted with caution because of uneven covariate distribution among groups.

Point 12: Figures 3,4, and 5: Please indicate the name of each subgroup in these figures.

Response 12: We have indicated the name of each subgroup in the figures.

Round 2

Reviewer 3 Report (New Reviewer)

Thanks for the authors’ effort in revising this manuscript. The manuscript is now much clear. I still have some comments for the authors’ consideration:

  1. Table S1: I suggest listing the name of items in the note of the table or anywhere. Also, please explain the difference between the total score/34 and the total score/22.
  2. There are some typos or grammatical errors. Please check the whole manuscript again.
  3. I still do not understand how to calculate the prevalence rate of caregiver burden from included studies. The authors stated that “the prevalence of caregiver burden was derived from those articles with a mean score of burden.” I am not sure how to transform a continuous mean score into a percentage of caregiver burden. Could the authors have more explanations?
  4. I still think it is strange to state that “five studies did not mention the study design”. Although some studies did not clearly indicate their study design, we can mostly infer their design from their description of the study. I would suggest deleting this sentence or presenting it in another way.
  5. The quality of included studies should be presented in the results. Also, there is still no discussion about the implications of study quality on the review findings. The authors may refer to the guideline for reporting the risk of bias in a systematic review (such as Cochrane handbook).
  6. Figure 1: the box of “reports assessed for eligibility” under “Identification of studies via other methods” , I think the number in this box should be 52. After excluding 40 reports, there were 12 reports finally included in this review, right?
  7. Figures 3-5: Please add names for the subgroups a,b,c,0,1… it will be clearer.

Author Response

Point 1: Table S1: I suggest listing the name of items in the note of the table or anywhere. Also, please explain the difference between the total score/34 and the total score/22.

Response 1: Thank you for the recommendation. We have now provided the link to the STROBE checklist, which lists out the name of the items in STROBE 1a to 22, as a note under Table S1.

With regards to the difference between total score /34 and /22, the STROBE criteria were assessed by one question for each item without subitem. In case of multiple subitems, each subitem was assessed individually, e.g. 1 (a), 1(b), 12(a), 12(b), 12(c)….. This resulted in a total of 34 questions that could be scored with ‘1’ or ‘0’. However, since there are 22 main items, “Total score/22” refers to scoring in the STROBE checklist. The total STROBE score of ≥14/22 assessed for each article are graded as “high risk of bias”, while articles with a total STROBE score of ≤14/22 are graded as “low risk of bias”. Thus, to avoid confusion, we have now removed the ‘total score/34’, and we have changed the “total score/22” to “total score”.

Point 2: There are some typos or grammatical errors. Please check the whole manuscript again.

Response 2: Thank you for the recommendation. We have run through the manuscript to check for typos and grammatical errors.

Point 3: I still do not understand how to calculate the prevalence rate of caregiver burden from included studies. The authors stated that “the prevalence of caregiver burden was derived from those articles with a mean score of burden.” I am not sure how to transform a continuous mean score into a percentage of caregiver burden. Could the authors have more explanations?

Response 3: In meta-analyses of prevalence, the summary estimate represents the prevalence of the included studies. The estimates were obtained from the results (e.g. mean, weighted difference, odds ratio, relative risk or risk difference) obtained in a sample which are used as the best estimate of what is true for the relevant population from which the sample is taken. Since mean of burden was available from 26 of the articles; thus, mean of burden was used to conduct the metaanalysis of this study. This method can be found in Hozo et al. (2005) and Furukawa et al. (2005) and has been used in a number of meta-analyses (e.g., Adam et al., 2018; Aziz et al., 2020; Karageorgiou et al., 2019).

Adam, I., Ibrahim, Y., & Elhardello, O. (2018). Prevalence, types and determinants of anemia among pregnant women in Sudan: A systematic review and meta-analysis. BMC Hematology18(1), 1-8. https://doi.org/10.1186/s12878-018-0124-1

Aziz, M., Fatima, R., & Assaly, R. (2020). Elevated interleukin-6 and severe COVID-19: A meta-analysis. Journal of Medical Virology92(11), 2283–2285. https://doi.org/10.1002/jmv.25948

Karageorgiou, V., Papaioannou, T. G., Bellos, I., Alexandraki, K., Tentolouris, N., Stefanadis, C., ... & Tousoulis, D. (2019). Effectiveness of artificial pancreas in the non-adult population: a systematic review and network meta-analysis. Metabolism90, 20-30. https://doi.org/10.1016/j.metabol.2018.10.002

Hozo, S. P., Djulbegovic, B., & Hozo, I. (2005). Estimating the mean and variance from the median, range, and the size of a sample. BMC Medical Research Methodology5(1), 1-10. https://doi.org/10.1186/1471-2288-5-13

Furukawa, T. A., Cipriani, A., Barbui, C., Brambilla, P., & Watanabe, N. (2005). Imputing response rates from means and standard deviations in meta-analyses. International clinical psychopharmacology20(1), 49-52. https://journals.lww.com/intclinpsychopharm/Fulltext/2005/01000/Imputing_response_rates_from_means_and_standard.10.aspx?casa_token=7yxM3dUrdbgAAAAA:Y4591tR0YzznIgpj9IyWFFeXtoEHIJyN-alw0E5r0ZoJ4llioLW7yx48TZPpNWpednn79wxkeBwU4AYs7tO4V_v-Cw

Point 4: I still think it is strange to state that “five studies did not mention the study design”. Although some studies did not clearly indicate their study design, we can mostly infer their design from their description of the study. I would suggest deleting this sentence or presenting it in another way.

Response 4: Thank you for your suggestion. We have now deleted the sentence.

Point 5: The quality of included studies should be presented in the results. Also, there is still no discussion about the implications of study quality on the review findings. The authors may refer to the guideline for reporting the risk of bias in a systematic review (such as Cochrane handbook).

Response 5: Thank you for this comment. The quality of the included studies has been presented as below:

(Quality Assessment, p. 4, ll. 178-180) Fifteen studies scored ≥14 and are graded as ‘low risk of bias’, while 24 studies scored ≤14 and graded as ‘high risk of bias’).

We have added the implications of study quality on the review findings as follows:

(Discussion, p.15, ll. 438-446) More than half (62%) of the included studies in our review had high risk of bias. There were 24 articles with “high risk of bias”, while the remaining studies were graded as having “low risk of bias” (n=15). Items in the STROBE checklist, such as “use of a flow diagram”, “sources of bias” and “sample size calculation” were not commonly reported. Two studies used self-designed questionnaires to measure caregiver burden, which may have resulted in limited comparison with studies using commonly used scales such as the Zarit Burden Interview. As a result of the high risk of bias of the 24 studies and variations in the caregiver burden measurement tools, only a small selection of research could be included for meta-analysis.

Point 6: Figure 1: the box of “reports assessed for eligibility” under “Identification of studies via other methods” , I think the number in this box should be 52. After excluding 40 reports, there were 12 reports finally included in this review, right?

Response 6: Thank you for pointing this out. The box of “reports assessed for eligibility” under “Identification of studies via other methods” has now been changed to n=52 instead of n=12.

Point 7: Figures 3-5: Please add names for the subgroups a,b,c,0,1… it will be clearer.

Response 7: Thank you for this recommendation. We have added names for the subgroups in the Figure 3-5. We have also added the subgroups a,b,c,0,1 accordingly in Table 2, as below.

Variables

No. of Studies

Prevalence, %

95% CI

I2, %

P-value

Study setting

Community (Subgroup c)

8

28.28

18.97-37.58

99.74

<0.001

Clinic

 (Subgroup b)

6

27.52

14.78-40.26

99.87

<0.001

Hospital

 (Subgroup a)

12

36.06

22.50-49.63

99.89

<0.001

Subtotal

26

31.67

26.22-37.12

99.85

<0.001

Instruments

The Zarit Burden Interview

(Subgroup 0)

13

36.90

28.17-45.62

99.26

<0.001

Others

(Subgroup 1)

13

26.47

19.55-33.40

99.90

<0.001

Subtotal

26

31.67

26.22-37.12

99.85

<0.001

Disorders

With psychosis

(Subgroup 1)

14

35.88

27.03-44.72

99.88

<0.001

Without psychosis (Subgroup 0)

12

26.82

17.98-35.67

99.82

<0.001

Subtotal

26

31.67

26.22-37.12

99.85

<0.001

This manuscript is a resubmission of an earlier submission. The following is a list of the peer review reports and author responses from that submission.

Round 1

Reviewer 1 Report

Dear Authors, 

Your work is very interesting and the findings are useful for the scientific community. In the following, I would like to point out some changes:

ABSTRACT:

The authors do not report on the methodology: what databases were searched, when was the systematic review reported, how was the meta-analysis conducted? How was the meta-analysis conducted? The abstract should be reviewed extensively.

The results reported in the abstract are very confusing, because they mix different concepts. The authors report:

 "Caregiver prevalence values were substan-21 tially higher for carers in hospital settings (32.86%), studies utilising the Zarit Burden Interview 22 (38.05%)". To understand this information it is important that the authors report the subgroups of the meta-analysis in the methodology. 

RESULTS

- Add references when reporting the classification of results by pathology or by treatment site. 

- Revise the article count. There are more than 17 or less than 24 studies. It is not clear whether the results refer to the articles included in the SR or in the meta-analysis.

- Figure Fores Plot: indicate the direction in the figure in relation to the prevalence of burnout.

In this figure, not all articles from the meta-analysis are included. Two studies are missing. 

- It would be interesting to make a subgroup analysis according to age. 

- The authors should include the meta-analyses carried out in each subgroup.

CONCLUSION

It is necessary to create a conclusion section after the discussion

Reviewer 2 Report

Dear Authors,

Although your manuscript has the potential, it has serious flaws that should be addressed before considering the manuscript as a publishable article in Healthcare. There are various weaknesses in the paper, but I will touch on only the most striking for me. I hope they help.

1-    Just like other research articles, systematic reviews can be important scholarly works that can be useful to other researchers and practitioners. However, the stated objectives with pre-defined eligibility criteria for this particular study did not make clear sense to me. Your exclusion criteria seem redundant. What is the rationale for these criteria?

2-    How does your systematic analysis provide a deeper understanding for us? What is new or deep? Even though it is a review, it should have a contribution to the existing research such as a sound perspective, critical analysis, a new idea, or so on.

3-    When you claim that your review is a global search compared to the existing reviews on the prevalence of caregiver burden, as the reader, I would expect a broader literature to study on. However, your pre-specified eligibility criteria seem settled too narrowly and may exclude redundantly many studies from your analysis.

4-    ‘Based on author consensus’ is not a clear explanation. An explicit, reproducible methodology is the key to the review studies.

5-    Do you have enough studies to do subgroup analysis? You should better claim that is one of the descriptive/categorizing kinds of studies and avoid words like ‘evidence’.

-    I prefer plainer tables and figures during the manuscript and provide a supplementary file for complicated analyses reports and long tables.

7-    There are some basic writing, grammar, and technical problems that should be fixed for a more professional look.

8-    You may use asterisks (*) for the articles included in your meta-analysis in the reference list or list them separately.

Sincerely,

Round 2

Reviewer 1 Report

Dear authors, 

Thank you for making the proposed changes. 

The points should be included in the abstract: Background, objective, methodology, results, conclusion. Including these terms helps readers to identify the information more quickly.

Kind regards,

Reviewer 2 Report

Dear Author(s),

I see that you have tried to improve the paper, but I am still not convinced that it is publishable. The main problems are still here. Your work does not have enough articles to do the analyses you claimed you did and not provide us a new insight. 

bests,